

# Integrated analysis reveals five potential ceRNA biomarkers in human lung adenocarcinoma

Yu Liu, Deyao Xie, Zhifeng He and Liangcheng Zheng

Department of Thoracic Surgery, The First Affiliated Hospital of Wenzhou Medical University, Wenzhou, Zhejiang, China

## ABSTRACT

**Background:** Competing endogenous RNAs (ceRNAs) are a newly identified type of regulatory RNA. Accumulating evidence suggests that ceRNAs play an important role in the pathogenesis of diseases such as cancer. Thus, ceRNA dysregulation may represent an important molecular mechanism underlying cancer progression and poor prognosis. In this study, we aimed to identify ceRNAs that may serve as potential biomarkers for early diagnosis of lung adenocarcinoma (LUAD).

**Methods:** We performed differential gene expression analysis on TCGA-LUAD datasets to identify differentially expressed (DE) mRNAs, lncRNAs, and miRNAs at different tumor stages. Based on the ceRNA hypothesis and considering the synergistic or feedback regulation of ceRNAs, a lncRNA–miRNA–mRNA network was constructed. Functional analysis was performed using gene ontology term and KEGG pathway enrichment analysis and KOBAS 2.0 software. Transcription factor (TF) analysis was carried out to identify direct targets of the TFs associated with LUAD prognosis. Identified DE genes were validated using gene expression omnibus (GEO) datasets.

**Results:** Based on analysis of TCGA-LUAD datasets, we obtained 2,610 DE mRNAs, 915 lncRNAs, and 125 miRNAs that were common to different tumor stages ($|\log_2(\text{Fold change})| \geq 1$, false discovery rate < 0.01), respectively. Functional analysis showed that the aberrantly expressed mRNAs were closely related to tumor development. Survival analyses of the constructed ceRNA network modules demonstrated that five of them exhibit prognostic significance. The five ceRNA interaction modules contained one lncRNA (FENDRR), three mRNAs (EPAS1, FOXF1, and EDNRB), and four miRNAs (hsa-miR-148a, hsa-miR-195, hsa-miR-196b, and hsa-miR-301b). The aberrant expression of one lncRNA and three mRNAs was verified in the LUAD GEO dataset. Transcription factor analysis demonstrated that EPAS1 directly targeted 13 DE mRNAs.

**Conclusion:** Our observations indicate that lncRNA-related ceRNAs and TFs play an important role in LUAD. The present study provides novel insights into the molecular mechanisms underlying LUAD pathogenesis. Furthermore, our study facilitates the identification of potential biomarkers for the early diagnosis and prognosis of LUAD and therapeutic targets for its treatment.

Corresponding author
Liangcheng Zheng,
zlc201810@126.com

## INTRODUCTION

Lung cancer, which is one of the most common malignancies worldwide, poses a serious threat to human health and life (*Jemal et al., 2011*; *Torre et al., 2015*). In recent years, there has been an increase in the morbidity of lung cancer owing to deterioration of the environment (*Cui et al., 2017*; *David et al., 2016*; *Haugen et al., 2000*; *Jin et al., 2014*; *Kooperstein, Schifrin & Leahy, 1965*; *Weiss, 1983*). Furthermore, lung cancer is difficult to diagnose early and associated with poor prognosis and low survival rate (*Bernaudin, 2010*; *Hardavella, George & Sethi, 2016*; *Onganer, Seckl & Djamgoz, 2005*; *Prim et al., 2010*; *Reinmuth et al., 2014*; *Tamura et al., 2015*). The prognosis of patients with lung cancer is not only related to its biological factors and immune status, but is also closely related with the therapeutic factors (*Fan, Zhang & Liu, 2014*; *Prim et al., 2010*; *Ringbaek et al., 1999*; *Truong, Viswanathan & Erasmus, 2011*); association between multiple factors leads to complex changes in the disease and makes prognosis difficult. Tumor staging is an important factor in determining prognosis, including survival in cancer patients (*Gong et al., 2017*; *Li et al., 2017*). Studies have shown that patients with late stage lung cancer display extremely poor prognosis, with invasion and metastasis, and additionally respond poorly to clinical treatment (*Mattern, Koomagi & Volm, 2002*; *Prim et al., 2010*). Therefore, early detection of cancer should enable administration of effective treatment and considerably improve patient prognosis. Although the prognosis of non-small cell lung cancer (NSCLC) has been greatly improved following the introduction of multidisciplinary treatment, the prognosis of lung cancer patients and treatment efficiency remain poor (*Fan, Zhang & Liu, 2014*; *Huang et al., 2012*; *Miura et al., 1996*; *Pujol et al., 1994*); therefore, the identification of biological markers and therapeutic targets for the early detection and treatment of recurrent lung tumor is critical.

Competing endogenous RNAs (ceRNAs) are a group of regulatory RNA molecules that compete with other RNA molecules to bind specific miRNAs, thereby regulating target gene expression (*Phelps et al., 2016*; *Shukla, Singh & Barik, 2011*; *Tay et al., 2011*). Many studies have suggested that miRNA-mediated ceRNA regulatory mechanisms play crucial roles in tumor occurrence and development (*Bartel, 2009*; *Hansen et al., 2013*; *Tay, Rinn & Pandolfi, 2014*). Therefore, the identification of ceRNA biomarkers related to lung cancer should enable the development of effective strategies for the diagnosis and treatment of lung cancer.

In this study, we aimed to identify differentially expressed genes (DEGs) associated with lung adenocarcinoma (LUAD) development and construct a ceRNA regulatory network using the TCGA-LUAD dataset. Further, we examined the relationship of the identified genes and ceRNA interaction modules with overall survival and prognosis. Through our study, we sought to gain new insights into the molecular mechanisms underlying LUAD and identify potential biomarkers for early diagnosis of this disease.

## MATERIALS AND METHODS

### Data source and preprocessing

Lung adenocarcinoma RNA-seq and lncRNA gene expression level three data were obtained from the database Cancer Genome Atlas (TCGA, https://tcga-data.nci.nih.gov/docs/publications/tcga/). After an initial filter (excluding samples with uncertain tumor stage),

569 samples were included for further study. Patient clinical data, including outcome and staging information, were also downloaded. Of the 569 samples, 510 were LUAD samples (168 with T1 stage, 276 with T2 stage, 47 with T3 stage, and 19 with T4 stage) and 59 were normal samples.

## Screening of genetically altered genes

Gene raw read counts were used to perform differential expression analysis with DESeq2 (v 1.18.1) (*Love, Huber & Anders, 2014*), a R package that uses a model based on the negative binomial distribution and is widely used for RNA-seq data differential analysis. Differentially expressed genes were defined by two criteria: false discovery rate (FDR) < 0.01 and fold change (FC) ≥ 2. In order to filter out the low-quality DEGs, we performed the following additional filter strategy. For mRNA and miRNA, we only retained DEGs that showed FPKM or RPKM larger than the threshold of one for at least 10% samples; for lncRNA DEGs, the threshold was set to 0.1. We then compared the normal tissue sample with different tumor stage samples (T1, T2, T3, and T4) to identify common DEGs for further analysis.

## Construction of lncRNA-associated ceRNA network (gene-set construction)

Based on the relationship between lncRNA, mRNA, miRNA, a ceRNA network was constructed using the following steps: 1) Both up-regulated and down-regulated LUAD-specific RNAs were chosen to construct the ceRNA network; 2) miRanda software was used to predict the miRNA–lncRNA interactions, and interactions in starBase (*Li et al., 2014*) were also included; 3) miRNA–mRNA interactions in miRTarbase and starBase were used; 4) Correlation between LUAD-specific lncRNA and mRNA was calculated with WGCNA (*Zhang & Horvath, 2005*) corAndPvalue function and correlation pairs with coefficient > 0.8 and *p*-value < 0.05 were retained; 5) The hypergeometric method was used to test the enrichment significance of miRNAs between mRNA and lncRNA. The enrichment significance was calculated by using the following formula:

$$P = \sum_{i=c}^{\min(K,\, n)} \frac{\binom{K}{i}\binom{N-K}{n-i}}{\binom{N}{n}}$$

In the above formula, $N$ represents the total number of miRNAs, $K$ is the number of miRNAs targeting mRNA, $n$ is the number of miRNAs targeting lncRNA, and $c$ is the number of miRNAs shared by the mRNA and lncRNA. Only ceRNA modules with FDR < 0.05 were considered.

## Gene functional enrichment analysis

In order to investigate key mRNAs at molecular and functional level, gene ontology (GO) and Kyoto Encyclopedia of Genes and Genomes (KEGG) pathway function enrichment analysis was performed. GOseq, which uses Wallenius' noncentral hypergeometric

distribution model taking gene length bias into account, was used to perform GO enrichment analysis. Gene ontology terms with $p$-value less than 0.05 were considered significantly enriched. KOBAS 2.0 software was used to test statistical enrichment in KEGG pathways, and those with a Fisher's exact test $p$-value of less than 0.05 was considered to be significantly enriched (*Xie et al., 2011*).

## Transcription factor (TF) analysis

Co-expression networks of LUAD-mRNA expression data sets were generated using Genie3 with EPAS1 and FOXF1 as Transcription factors (TFs) of interest. A threshold of 0.005 was used to filter the co-expression network. Additionally, Spearman correlations between the TFs and LUAD-specific mRNAs were calculated, and TF-mRNA pairs with coefficient > 0.03 were retained as this value represents positive correlation. Using this procedure, we obtained a TF co-expression draft regulatory network. As the analysis was only based on co-expression, the results may include numerous indirect targets (such as downstream effects). To identify genes that are most likely the direct targets, we used RcisTarget (*Aibar et al., 2017*; *Imrichova et al., 2015*) to perform cis-regulatory motif analysis of each TF regulon. RcisTarget is an R package that identifies TF binding motifs that are over-represented in a gene list. We used the database, scoring 500 bp upstream of TSS, and only motifs with a Normalized Enrichment Score (NES) > 3.0 were considered associated with TFs of interest.

## Validation of gene expression with gene expression omnibus data

In order to validate the key differentially expressed (DE) molecules, we examined four datasets from among the gene expression omnibus (GEO) datasets. Of the selected datasets, the GSE10072 study contained 58 LUAD tumor and 49 non-tumor tissue mRNA expression data, GSE32863 contained 58 LUAD tumor and 58 non-tumor tissue mRNA expression data, GSE85716 contained six LUAD tumor and six non-tumor tissue lncRNA expression data, and GSE104854 contained three LUAD (pooled from nine tissues) and three non-tumor (pooled from nine tissues) expression data. GSE10072, GSE32863, and GSE85716 datasets, which were derived from microarray data, were analyzed with GEO2R online. The GSE104854 dataset, which was obtained by high-throughput sequencing, was analyzed with the DESeq2 R package based on read counts.

## Statistical analysis

To investigate the impact of the expression level of RNAs on the prognostic survival of patients, survival analysis was performed. In the analysis, Kaplan–Meier survival analyses and log-rank test were performed to study the relationship between RNA expression states (cutoff point: median value) and survival time. Univariate cox proportional hazards regression was applied to identify the RNAs associated with overall survival. In order to evaluate the potential of dysregulated ceRNA modules as biomarkers, the risk scoring classifier was constructed. For each ceRNA module, we calculated the risk score for multivariate survival analysis to determine the prognostic influence of the RNAs as a whole. All statistical analyses were performed using R version 3.4.3 software (*R Core Team, 2017*).

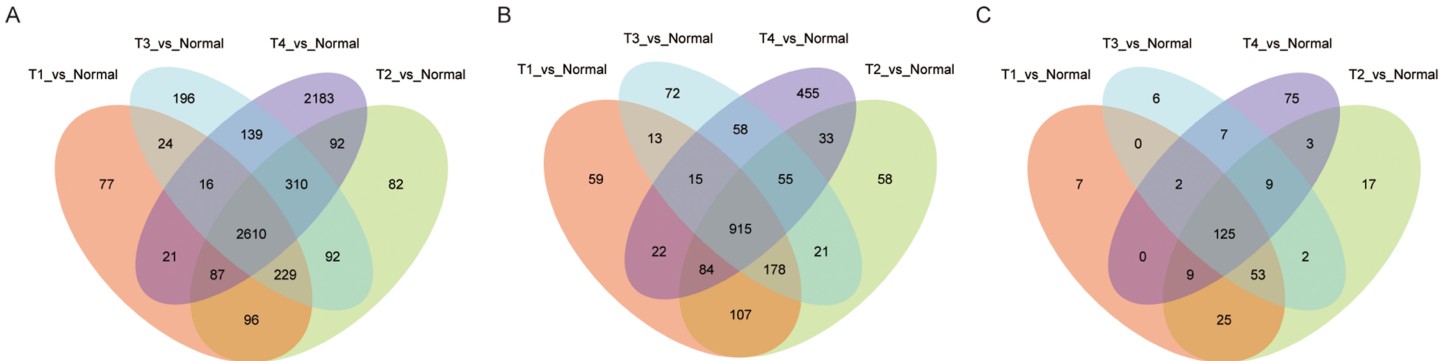

**Figure 1 Significantly expressed genes in LUAD.** (A) The Venn diagram of mRNA DEGs, the circle represents mRNA DEGs between normal tissue and tumor stage tissue; (B) the Venn diagram of lncRNA DEGs; (C) the Venn diagram of miRNA DEGs.

## RESULTS

### Differentially expressed RNAs in LUAD

Among the LUAD datasets, expression data for 19,669 mRNAs, 7,309 lncRNAs, and 1,882 miRNAs were extracted from TCGA and analyzed with R package (DESeq2). In this analysis, we compared adjacent normal lung tissue with lung cancer subtypes defined by pathology stage (T1-T4), respectively. When we combined these four groups and analyzed for DE RNAs, 2,160 mRNAs (1,527 up- and 1,083 down-regulated), 915 lncRNAs (662 up- and 253 down-regulated), and 125 miRNAs (73 up- and 52 down-regulated) showed consistently differential expression (FC ≥ 2, FDR < 0.01) (Figs. 1A–1C). Figs. 2A–2C shows the heatmap of the common DEGs; the lung tumor tissue samples could be easily distinguished from the adjacent non-tumor lung tissues from the heat map. Based on these data, DE mRNAs, lncRNAs, and miRNAs were selected for further analysis.

### Functional enrichment of mRNA DEGs

In order to understand the functions of aberrantly expressed genes in this study, a total of 2,610 aberrantly expressed genes were analyzed. We used GOseq for GO annotation and KOBAS 2.0, which includes the KEGG database, to classify and analyze the potential gene functions in the pathways. Our analysis revealed an enrichment of 2,879 GO terms and 30 KEGG pathways ($p$ value < 0.05) (Table S1). Gene ontology analyses were classified into three functional groups: molecular function group, biological process group, and cellular component group (Fig. 3). Enrichment of these DEGs represents a measure of significance of a function. As shown in Fig. 3, the most enriched GO biological process terms were "single-organism cellular process" and "single-multicellular organism process." Other significant GO terms included "cell proliferation," "cell adhesion," "system development," "cell migration," and "cell adhesion." The significant pathways identified by KEGG pathway analysis included viral carcinogenesis, systemic lupus erythematosus, cell adhesion molecules, and p53 signaling pathway (Fig. 4). Our analysis indicated that 2,610 dysregulated mRNAs were involved in signaling pathways related to environmental information processing and human diseases such as cancer.

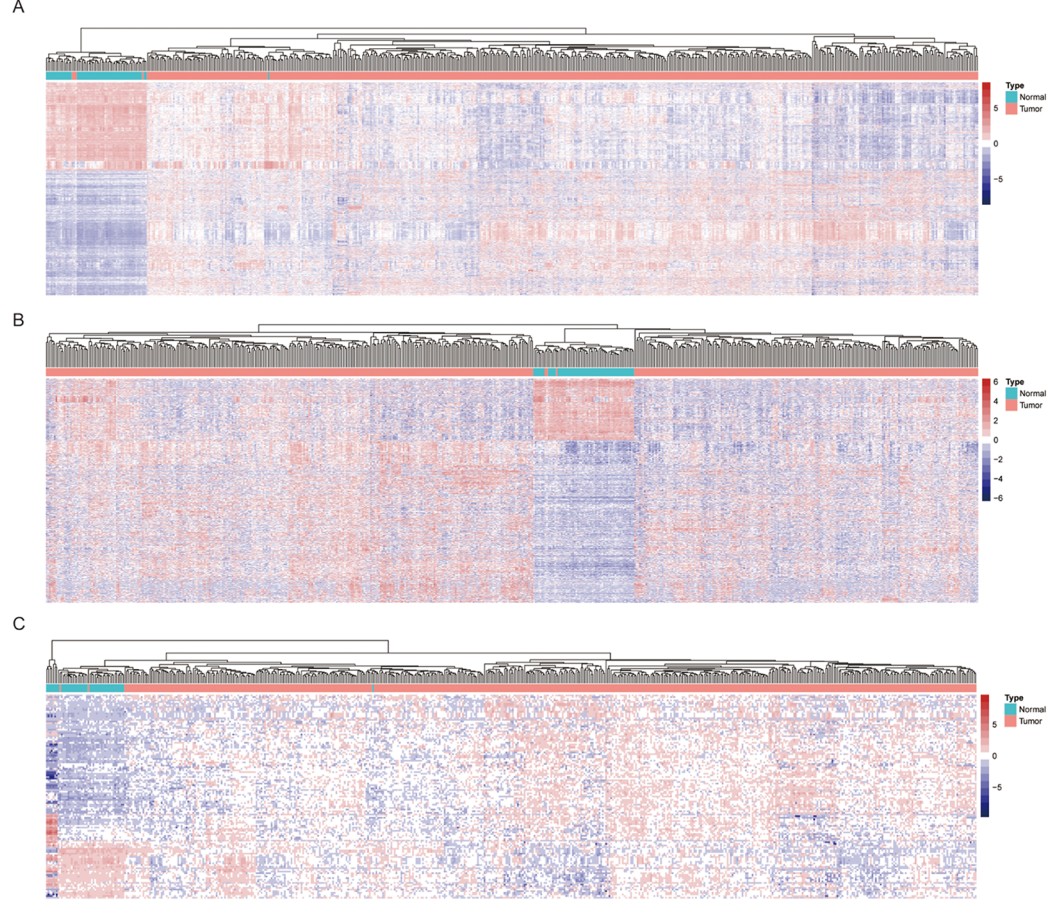

**Figure 2 Heatmap of significantly expressed common genes between different pathology stages of LUAD.** (A) mRNA DEGs; (B) lncRNA DEGs; (C) miRNA DEGs.

## Competing endogenous RNA network analysis and integrated ceRNA network construction

Based on the miRNAs and mRNAs described in Fig. 2, starBase v2.0 database and miRanda were used to predict miRNA-targeted RNAs. Then, combing miRNA–lncRNA interactions with the miRNA–mRNA interactions, an integrated lncRNA–miRNA–mRNA ceRNA network was established. As shown in Fig. 5, the network contained 107 interactions, and included 5 lncRNAs, 20 mRNAs, and 34 miRNAs (Table S2). After ranking the molecules according to their degree, FENDRR, which is a lncRNA, showed the highest degree (degree = 38). *FOXF1*, *EDNRB*, and *EPAS1* are all mRNA molecules, ranking the second, third, and fourth respectively. In other studies, *FENDRR* was demonstrated to be down-regulated in breast and gastric cancers and to participate in regulating tumor malignancy (*Li et al., 2018*; *Xu et al., 2014*). FOXF1 and EPAS1 are TFs that have been demonstrated to participate in tumor progression (*Putra et al., 2015*; *Tamura et al., 2014*). EDNRB is a G protein-coupled receptor that activates a phosphatidylinositol-calcium second messenger system, and is also closely related with cancer (*Chen et al., 2013*; *Wuttig et al., 2012*).

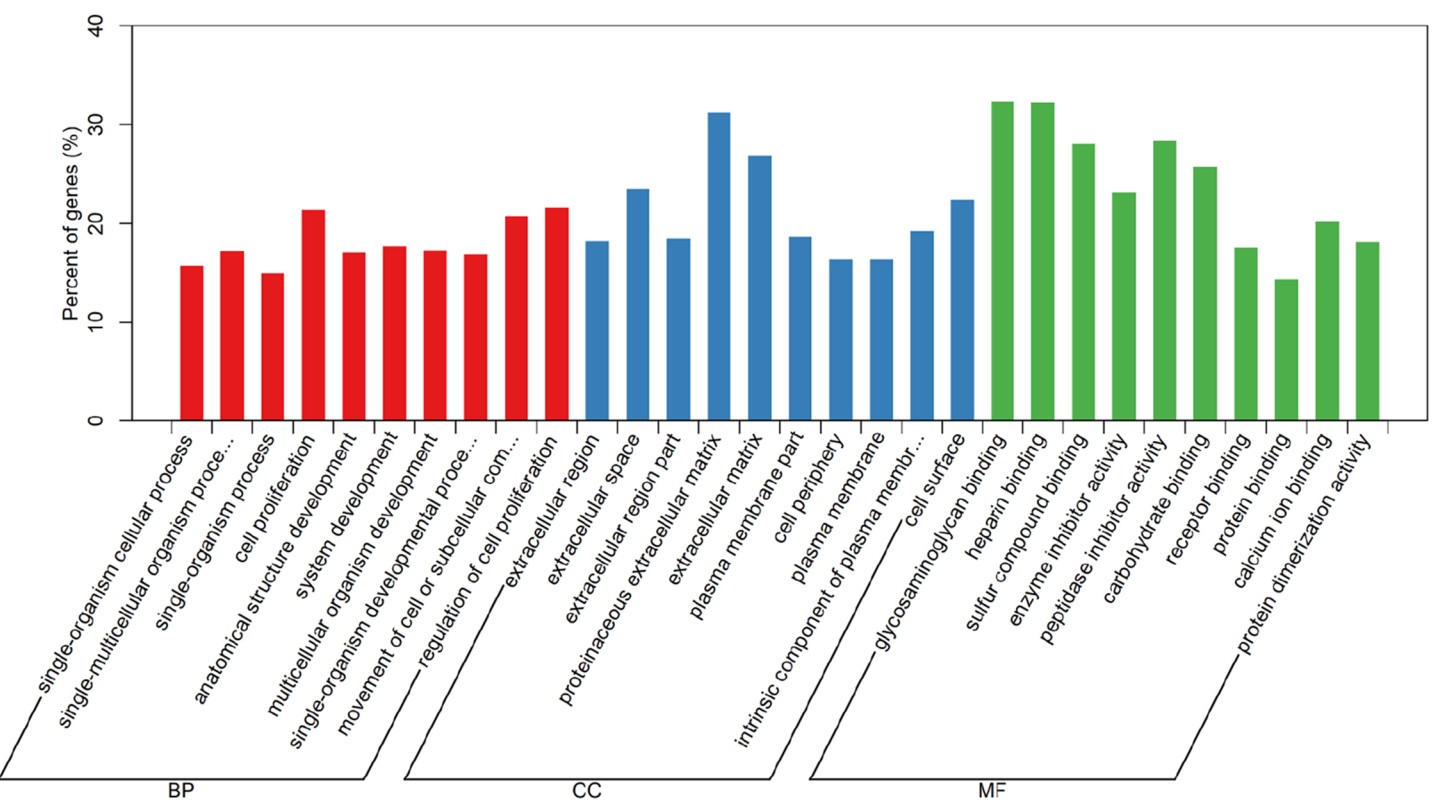

**Figure 3 Gene Ontology analysis and significant enriched GO terms of DEGs in LUAD.** GO analysis classified the DEGs into three groups (i.e., molecular function, biological process and cellular component).

## Key ceRNAs and their association with clinical features

In order to further explore the key role of ceRNAs in the occurrence and development of LUAD, we aimed to identify prognostic-specific ceRNAs. We calculated expression of each ceRNA and performed prognosis analyses (e.g. of overall survival time) to determine the survival-significant lncRNA–miRNA–mRNA interactions based on Cox analysis ($p$ value < 0.05). As presented in Fig. 6 and Table 1, univariate Cox regression analysis showed that EDNRB-FENDRR-hsa-mir-196b, EPAS1-FENDRR-hsa-mir-148a, FOXF1-FENDRR-hsa-mir-148a, FOXF1-FENDRR-hsa-mir-195, and FOXF1-FENDRR-hsa-mir-301b ceRNA interactions were associated with the overall survival of patients with LUAD. Kaplan–Meier survival curves indicated that EDNRB-FENDRR-hsa-mir-196b interaction was negatively correlated with overall survival, whereas EPAS1-FENDRR-hsa-mir-148a, FOXF1-FENDRR-hsa-mir-148a, FOXF1-FENDRR-hsa-mir-195, and FOXF1-FENDRR-hsa-mir-301b were positively correlated with survival.

## Transcription factor cis-target analysis

Transcription factors play important roles in the regulation of biological processes by binding enhancer or promoter regions of DNA, thereby activating or suppressing gene expression. This allows genes to be expressed in the right cell at the appropriate time. In this study, we found that EPAS1 and FOXF1 showed a high degree of representation

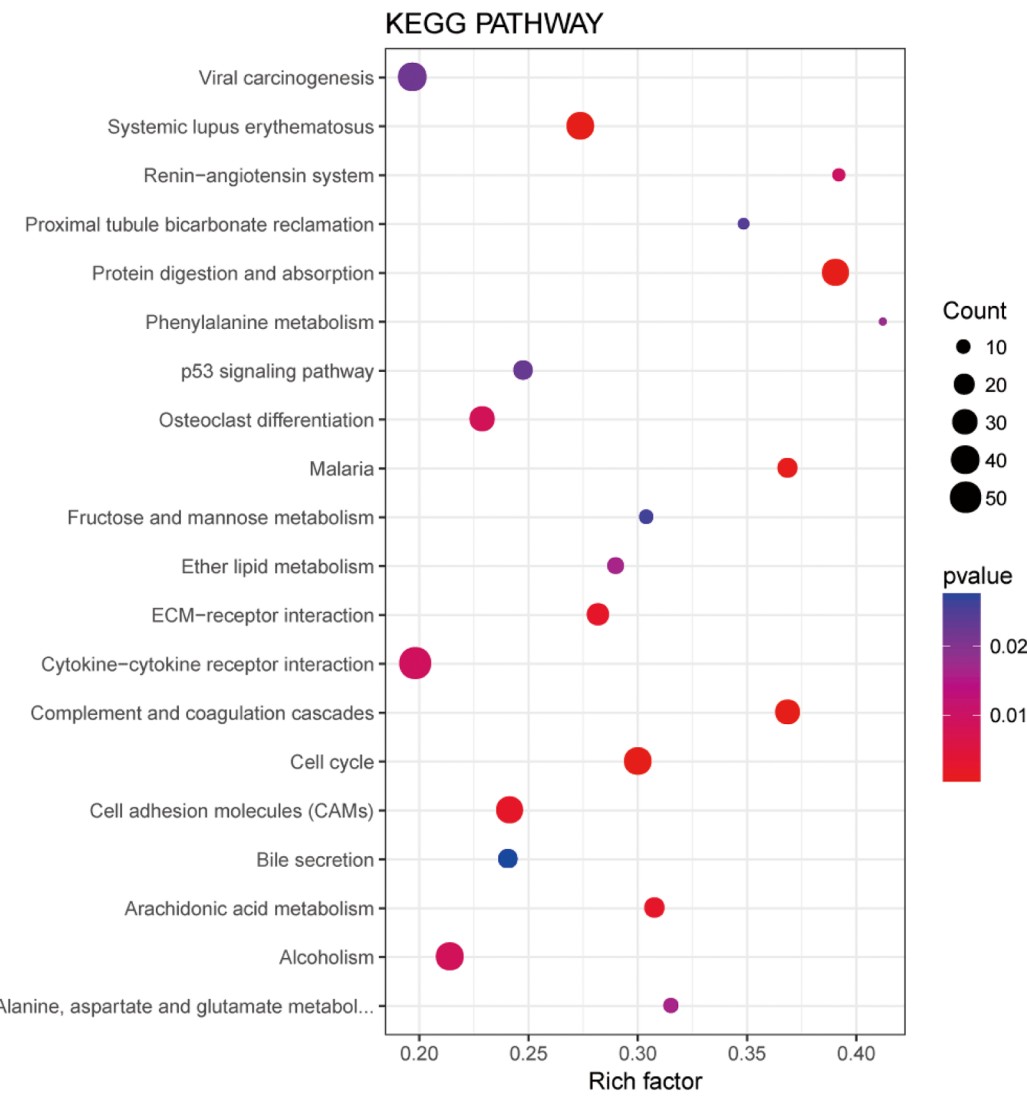

**Figure 4 Significantly enriched pathway terms in LUADs and 5ST.** The functional and signaling pathway enrichment were conducted using KOBAS 2.0 software.

in the ceRNA network. This finding suggests that these two TFs play an important role in LUAD and may be regulated by a ceRNA-based mechanism. In order to verify the direct targets of the two TFs, we analyzed them further. Co-expressed genes may be regulated by the same or similar transcription factors; however, they may include numerous indirect targets. Therefore, we performed binding motif enrichment analysis to identify direct target genes of the two TFs. In this study, we use GENIE3 to perform TF-genes co-expression analysis and RcisTarget to analyze enriched motifs in the TSS upstream of the gene-sets. As shown in Fig. 7, the *EPAS1* co-expressed gene-set was enriched for the "cisbp__M6212" motif (NES = 3.21); likely direct gene targets of EPAS1 include *ADRB2*, *CALCRL*, *EMP2*, *FHL1*, *GRIA1*, *LGI3*, *LIMS2*, *NCKAP5*, *RXFP1*, *SLC6A4*, *SMAD6*, *STX11*, and *TMEM100*. However, the *FOXF1* co-expressed gene-set did not show enrichment for any motifs.

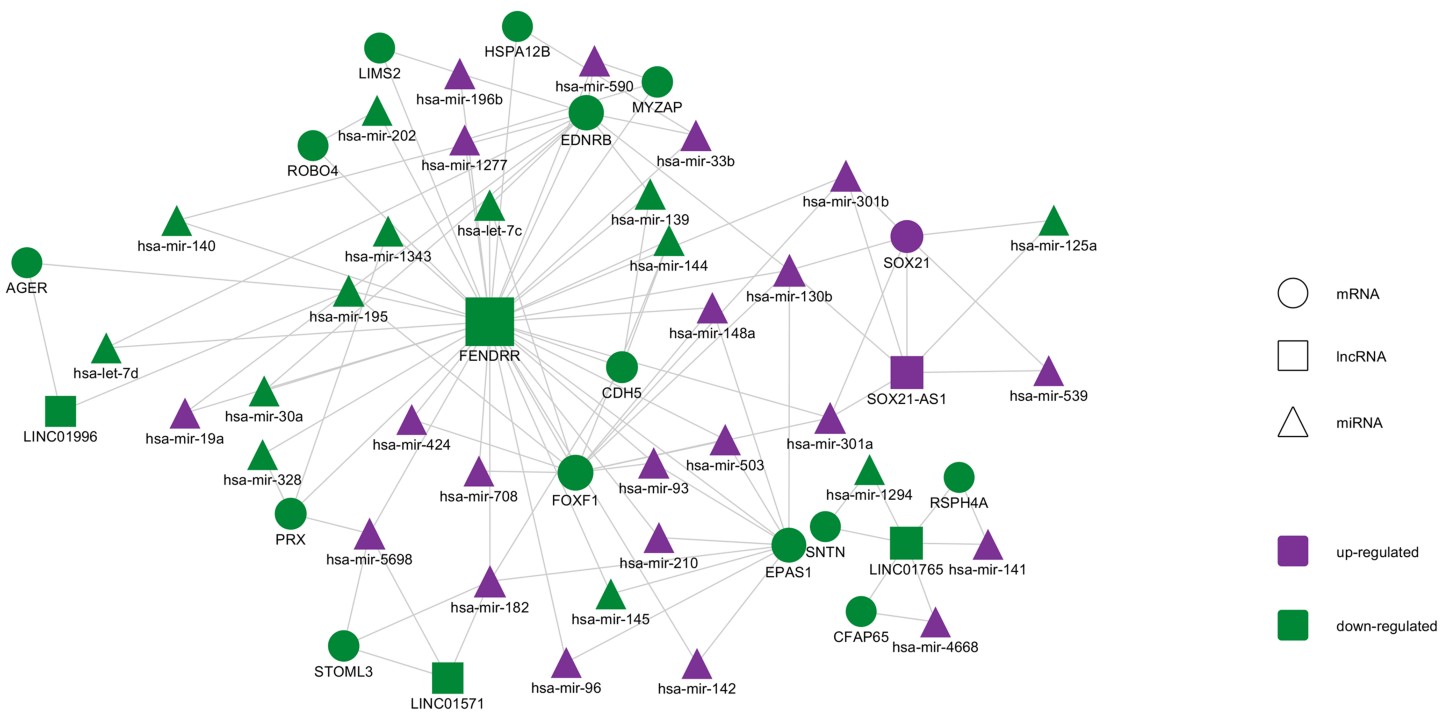

**Figure 5 The lncRNA–miRNA–mRNA ceRNA network constructed from DEGs.** Purple represents up-regulated genes and yellow represents down-regulated genes. Circles represent mRNA, rectangles represent lncRNA, and triangles represent miRNA.

## Validation of the expression of key RNA molecules with GEO data

Finally, four reported studies were screened out from the GEO to verify the differential expression of key mRNAs and lncRNAs in LUAD. GSE10072 and GSE32863 were used for *EPAS1*, *FOXF1*, and *EDNRB* mRNA verification. GSE85716 and GSE104854 were used for *FENDRR* differential expression. As shown in Fig. 8 and Table 2, all four molecules showed significantly lower levels of expression in tumor datasets, which is consistent with the results of our study.

## DISCUSSION

Non-small cell lung cancer, which mainly includes adenocarcinoma and squamous cell carcinoma, accounts for about one-sixth of cancer deaths in the global population (*Jemal et al., 2011*; *Torre et al., 2015*). Despite numerous advances in the treatment of lung cancer, the prognosis of this disease remains poor, and the 5-year overall survival rate is only 11% (*Huang et al., 2012*; *Miura et al., 1996*). Therefore, elucidation of the underlying mechanisms is indispensable for diagnosing, preventing, and treating NSCLC.

With the development of high-throughput sequencing technology, accumulating omics data reveal that the occurrence and development of the disease is closely related to multiple factors, including genomic mutations, expression profiles, copy number variants, and structural variations (*Balbin et al., 2013*; *Kikutake & Yahara, 2016*; *Kim et al., 2015*; *Yan et al., 2017*). In recent years, several studies have found that multiple RNA molecules are closely related to tumor progression (*Chen et al., 2018*; *He et al., 2018*;

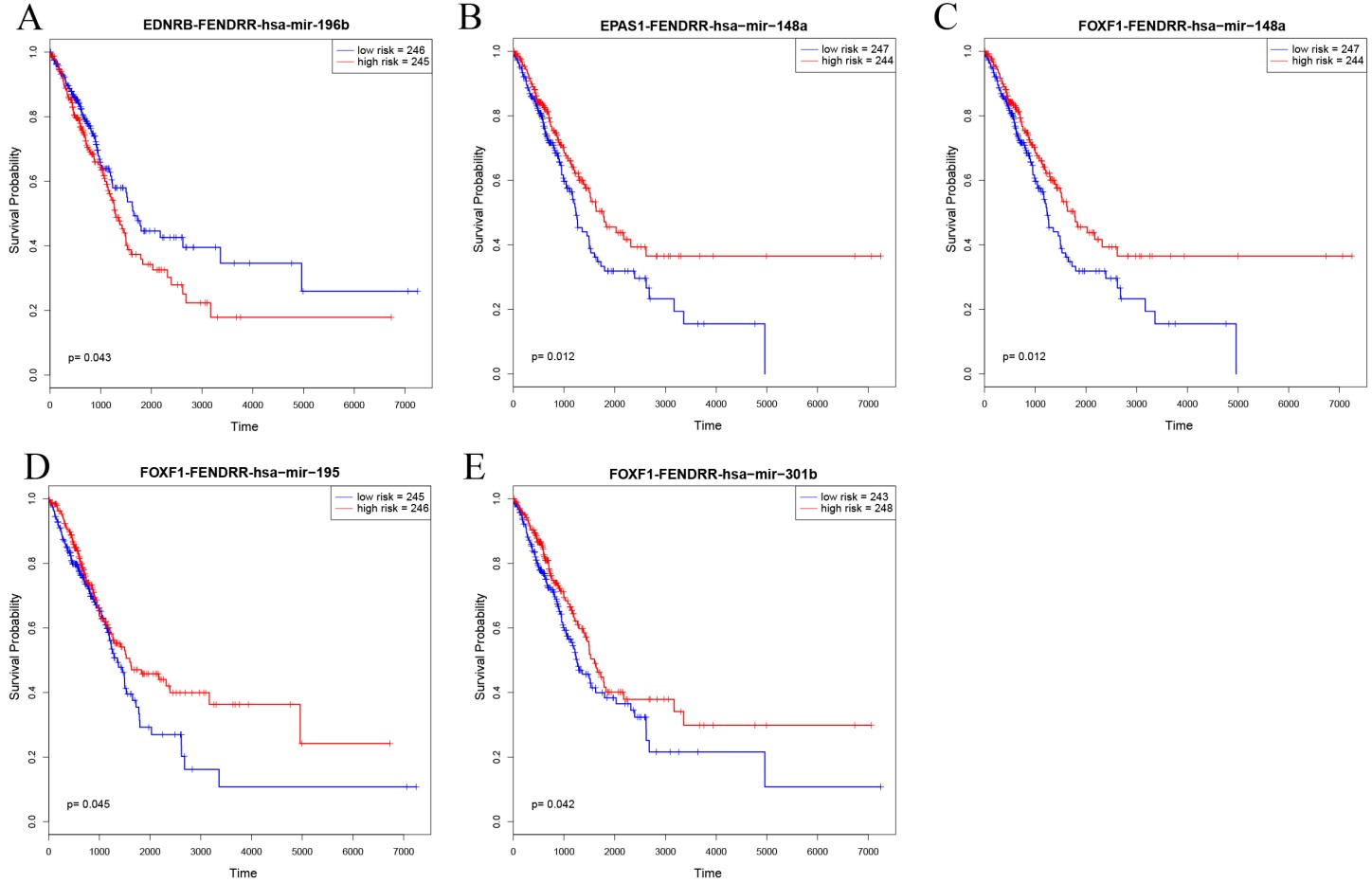

**Figure 6 Kaplan–Meier survival curves for five ceRNA pairs associated with overall survival.** (A) EDNRB-FENDRR-hsa-mir-196b. (B) EPAS1-FENDRR-hsa-mir-148a. (C) FOXF1-FENDRR-hsa-mir-148a. (D) FOXF1-FENDRR-hsa-mir-195. (E) FOXF1-FENDRR-hsa-mir-301b.

**Table 1 Gene fold change in five ceRNA pairs.**

| Gene | T1_vs_Normal | | T2_vs_Normal | | T3_vs_Normal | | T3_vs_Normal | |
|---|---|---|---|---|---|---|---|---|
| | log2FC | FDR | log2FC | FDR | log2FC | FDR | log2FC | FDR |
| EPAS1 | −2.64 | 5.65E-97 | −2.76 | 1.1E-127 | −2.49 | 1.3E-64 | −2.61 | 9.45E-41 |
| FOXF1 | −2.44 | 6.68E-55 | −2.96 | 2E-126 | −2.58 | 1.2E-45 | −2.63 | 8.73E-30 |
| EDNRB | −3.27 | 2.09E-78 | −3.75 | 9.2E-123 | −3.38 | 3.4E-60 | −2.94 | 8.47E-38 |
| FENDRR | −3.48 | 2.85E-54 | −4.37 | 2.7E-112 | −3.78 | 1.62E-42 | −3.89 | 1.94E-42 |
| hsa-mir-148a | 1.51 | 1.31E-24 | 1.46 | 1.51E-23 | 1.45 | 4.7E-11 | 1.51 | 1.07E-07 |
| hsa-mir-195 | −1.98 | 6.47E-37 | −2.25 | 5.03E-68 | −2.28 | 2.9E-30 | −2.08 | 6.23E-13 |
| hsa-mir-196b | 3.00 | 1.59E-21 | 4.42 | 7.94E-35 | 2.98 | 5.6E-15 | 4.21 | 1.08E-20 |
| hsa-mir-301b | 3.48 | 2.62E-22 | 4.14 | 7.95E-28 | 2.94 | 9.5E-15 | 3.02 | 1.82E-05 |

*Men, Liu & Ren, 2018*; *Shang et al., 2017*; *Song et al., 2018*; *Tang et al., 2017*; *Wang et al., 2018*; *Yu et al., 2017*). For example, lncRNA-DANCR enhances the stemness feature of cancer cells to increase the degree of malignancy of lung cancer (*Lu et al., 2018*;

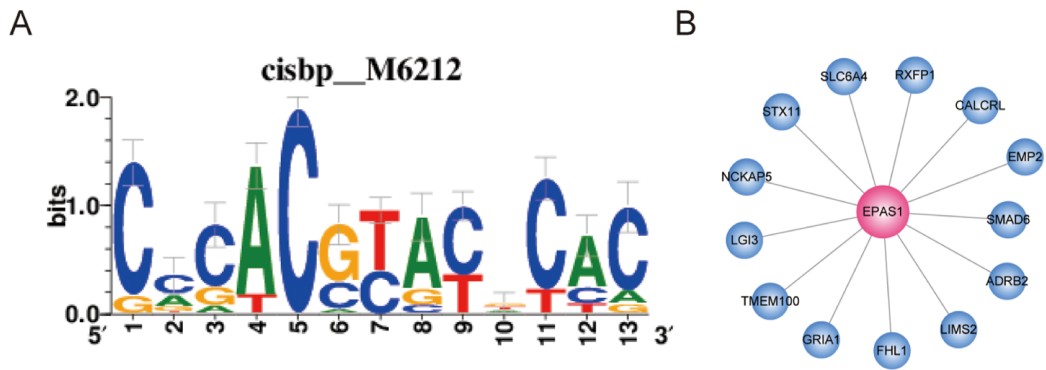

**Figure 7 Co-expressed genes analysis for EPAS1.** (A) EPAS1 co-expressed gene-set was enriched for motif "cisbp__M6212." (B) EPAS1-regulated genes in LUAD.

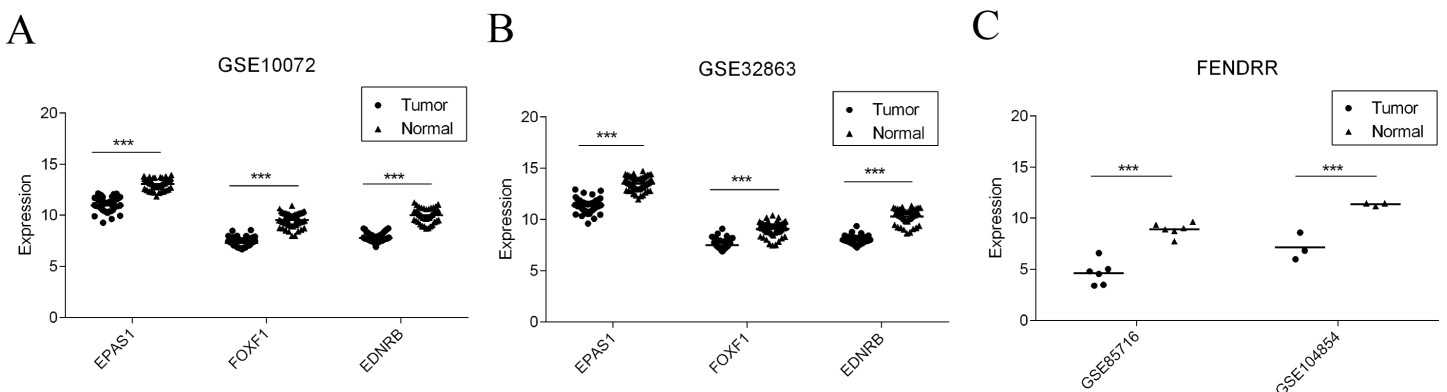

**Figure 8 Validation of gene expression in GEO dataset.** (A) EPAS1, FOXF1, and EDNRB expression level in GSE10072. (B) EPAS1, FOXF1 and EDNRB expression level in GSE32863. (C) FENDRR expression level in GSE85716 and GSE104854.

**Table 2 EPAS1, FOXF1, EDNRB and FENDRR expression in GEO dataset.**

| Study | mRNA | | | | | | lncRNA | |
|---|---|---|---|---|---|---|---|---|
| | EPAS1 | | FOXF1 | | EDNRB | | FENDRR | |
| | log2FC | FDR | log2FC | FDR | log2FC | FDR | log2FC | FDR |
| GSE10072 | −2.087 | 3.08E-33 | −2.228 | 2.17E-35 | −2.299 | 4.30E-35 | – | – |
| GSE32863 | −2.152 | 3.05E-33 | −1.616 | 2.69E-28 | −2.385 | 2.35E-44 | – | – |
| GSE85716 | – | – | – | – | – | – | −4.25009 | 0.000343 |
| GSE104854 | – | – | – | – | – | – | −4.12757 | 3.96E-08 |

*Zhen et al., 2018*); miRNA-221 was reported to regulate the process of tumor differentiation, proliferation, migration, and invasion (*Lv et al., 2014*; *Yin, Xu & Li, 2017*). These RNA molecules represent biological markers for the prognosis of lung cancer. In this study, we found that multiple novel ceRNA molecules can be used as novel prognostic markers; further, we examined the putative molecular mechanisms underlying the development of tumors.

miRNAs modulate the expression of target genes by regulating the transcription and stability of their mRNAs or lncRNAs. Investigation of the lncRNA-mRNA co-expression network is important for analysis of the function and regulatory mechanisms of lncRNAs (*Fang, Wang & Li, 2018*). In this study, analysis showed that a number of mRNAs, lncRNAs, and miRNAs that were DE were common at each stage of lung cancer. Gene ontology and KEGG analysis showed that the DEGs were mainly involved in "single-organism cellular process," "single-multicellular organism process," "cell proliferation," "cell adhesion," "system development," "cell migration," and "cell adhesion biological processes." Further analysis revealed that the up-regulated genes were involved in the mitotic cell cycle and nuclear division process (Table S3), while down-regulated genes were involved in vasculature development, cell motility, cell migration, and cell proliferation (Table S4). Kyoto encyclopedia of genes and genomes analysis found that the up-regulated genes were mainly involved in cancer and other related signaling pathways such as the p53 signaling pathway.

Survival curve analysis reveals the correlation between gene expression and patient prognosis. Our study found that the five most important ceRNA-related RNA molecules were significantly correlated with the survival rate of patients with lung cancer ($p < 0.01$), and most of the genes were highly correlated with cancer. The five ceRNA interactions consisted of one lncRNA, three mRNAs, and four miRNAs. The lncRNA, FENDRR (*Grote et al., 2013*), which is produced from a spliced long non-coding RNA transcribed bidirectionally with FOXF1 on the opposite strand, is down-regulated in both breast cancer and gastric cancer. In breast cancer, lower expression level of FENDRR was associated with shorter overall survival and progression-free survival in breast cancer patients. FENDRR knockdown promoted breast cancer cell proliferation and migration, and suppressed cell apoptosis; in contrast, its overexpression inhibited tumor growth in a xenograft model (*Li et al., 2018*). FENDRR was also reported to be down-regulated in gastric cancer cell lines and tissues. Further, FENDRR expression was negatively correlated with tumor metastasis and stage (*Xu et al., 2014*). In our study, FENDRR was also down-regulated in LUAD, and showed the highest degree in ceRNA network, indicating its important role in LUAD. The three mRNAs, *FOXF1*, *EDNRB*, and *EPAS1*, ranked 2nd, 3rd, and 4th, respectively in the ceRNA network. FOXF1 and EPAS1 are both TFs that have been associated with tumor malignancy (*Putra et al., 2015*; *Tamura et al., 2014*). Our TF analysis indicated that EPAS1 directly regulates 13 molecules associated with LUAD, namely ADRB2, CALCRL, EMP2, FHL1, GRIA1, LGI3, LIMS2, NCKAP5, RXFP1, SLC6A4, SMAD6, STX11, and TMEM100. EDNRB, a G protein-coupled receptor, is also closely related with tumor (*Chen et al., 2013*; *Wuttig et al., 2012*). Our findings suggest that these molecules represent potential cancer-related biomarkers.

## CONCLUSIONS

In summary, in the present study, we comprehensively analyzed the expression of key RNA molecules in the progression of lung cancer and identified key RNA molecules (including mRNA/lncRNA/miRNA) that were abnormally expressed in different stages of this disease. Further, using the above-mentioned key RNA molecules to construct a ceRNA–gene interaction network, it was found that several FOXF1- and EDNRB-related

ceRNA molecules play an important role in the development and progression of lung cancer, and can affect the prognosis of this disease. This series of related ceRNA molecules is expected to provide a novel basis for diagnosis and evaluation of prognosis, and represents potential new targets for the treatment of lung cancer.

## ACKNOWLEDGEMENTS

The authors would like to thank TCGA for providing the data and Mrs. Gu JY for her technical assistance.

### Funding

The study was supported by the Natural Science Foundation of Zhejiang Province (LY17H160048). The funders had no role in study design, data collection and analysis, decision to publish, or preparation of the manuscript.

### Grant Disclosures

The following grant information was disclosed by the authors:
Natural Science Foundation of Zhejiang Province: LY17H160048.

### Competing Interests

The authors declare that they have no competing interests.

### Author Contributions

- Yu Liu conceived and designed the experiments, performed the experiments, analyzed the data, contributed reagents/materials/analysis tools, prepared figures and/or tables, authored or reviewed drafts of the paper, approved the final draft.
- Deyao Xie performed the experiments, analyzed the data, prepared figures and/or tables.
- Zhifeng He performed the experiments, analyzed the data, prepared figures and/or tables.
- Liangcheng Zheng conceived and designed the experiments, contributed reagents/materials/analysis tools, authored or reviewed drafts of the paper, approved the final draft.

### Data Availability

The origin RNAseq data used in our study were all downloaded from the TCGA data portal: https://portal.gdc.cancer.gov/;
GEO data link of mRNA: GSE10072; GSE32863;
GEO data link of lncRNA: GSE85716; GSE104854.

### Supplemental Information

Supplemental information for this article can be found online at http://dx.doi.org/10.7717/peerj.6694#supplemental-information.

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
