# Peer review of "Integrated analysis reveals five potential ceRNA biomarkers in human lung adenocarcinoma"

_PeerJ, doi:10.7717/peerj.6694_

## Round 0.1 · original submission · Major Revisions

The reviewers have raised a number of points which we believe would improve the manuscript. If you are able to fully address these points, we would encourage you to submit a revised manuscript.

Reviewer 1 ·

Basic reporting

no comment

Experimental design

no comment

Validity of the findings

Speculation is welcome, but should be identified by molecular biology experiments, at least real-time fluorescent quantitative PCR

Additional comments

Authors identified differentially expressed (DE) mRNAs, lncRNAs and miRNAs at different tumor stages based on TCGA database, and then constructed a lncRNA-miRNA-mRNA network based on ceRNA hypothesis.They found five potential ceRNA biomarkers in human lung adenocarcinoma. Further confirmation was performed based GEO database. These results were meaningful, however some questions need to be answered.
1. In the legend of Figure 5, yellow represents down-regulated genes?
2. For survival analysis in Figure 6, which RNA (mRNA, lncRNA or hsa-mir-196b) expression was cut off based on median value?
3. The manuscript needs language polishing.

Reviewer 2 ·

Basic reporting

In this manuscript, Liy et al., aim to characterize the expression of competing endogenous RNAs (ceRNAs) in lung adenocarcinoma (LUAD). Overall, the manuscript is well constructed. However, there are a few instances where precise experimental details are not clearly presented . A few minor revisions could help strengthen the quality of the manuscript:

Experimental design

1) In figure 6, the authors should clarify the nature of the two arms of each of the Kaplan-Meier curves. Firstly, do “low-” and “high-risk” refer to the expression of the miRNA-mRNA combinations? Secondly, it is unclear what exactly each “ceRNA pair” in this experiment refers to. Are the expressions of each of 3 RNA molecules in each “pair” correlated?

2) Related to the above, in the discussion, the authors note that low expression of FENDRR has been associated with shorter overall survival and progression free survival in breast cancer. Was a similar correlation observed by the authors in this study? Figure 6B suggests the opposite might be the case in LUAD

3) The presence of a consensus motif for genes co-expressed with EPAS1 (figure 7) is intriguing. It would be curious if the existence of such a motif has been previously documented in the literature.

4) It is unclear what is meant in line 198, where the authors state, FENDRR is a lncRNA which showed the highest “degree”. Does this refer to the level of differential expression, or the level of inter-connectivity with other differentially expressed genes?

Validity of the findings

For the most part, global gene expression analysis as conducted in this study can be an extremely helpful resource. However, an issue in this manuscript is the over-interpretation of these results. As such, I would urge the authors refrain from making broad conclusive statements. For example, in line 219, they claim “these results suggested that these ceRNAs likely play an important role during lung cancer development”. Similarly, in line 302 “our study indicates that lncRNA related ceRNAs and TFs play important roles in LUAD”. Such sweeping conclusions without any functional data are a stretch and should be avoided.

Reviewer 3 ·

Basic reporting

The work presented is a leading bioinformatics work that honours a peer-reviewed journal like Peer J. Based on a classic approach, but innovative when targeted molecules, the authors do a remarkable job. One of the difficulties of in silico approaches is to see the limitations of approaches and analyses. Here, there are no such problems.
The state of the art is necessary and sufficient, without articles in excess for convenience. The whole process is well brought into context.
The figures could be improved; some require too much zoom to be readable.
The ms. is self-contained with relevancy results to hypotheses.

Experimental design

The design is rigorous and well balanced.
Research is well defined, relevant & meaningful. It would be difficult to ask for additional work, as it is already quite complete.
Methods are described with sufficient detail, and could be replicate.
My only minor question can be if the ceRNA network constructed from DEGs is not too dependent of some hubs.

Validity of the findings

The design is simple and strict, so the results are of similar quality. The Conclusion are well stated, linked to original research question & limited to supporting results.

Additional comments

a simple and pleasant article to read. Well detailed and rigorous.

Reviewer 4 ·

Basic reporting

No Comments

Experimental design

NO comments

Validity of the findings

No comments

Additional comments

The manuscript by Liu et al titled "Integrated analysis reveals five potential ceRNA biomarkers in human lung adenocarcinoma uses online available datasets and online tools to find differentially expressed mRNAs, lncRNA and miRNA at different stages of lung cancer. The paper is very very preliminary and does not present any thing significant which can published in scientific journal. I do not recommend its publication.

---

## Round 0.2 · accepted · Accept

The author made a lot of efforts to improve their manuscript. I think their study can be published.

# Reviewer 2 ·

Basic reporting

The author's have addressed comments from my prior review. I have no further comments

Experimental design

The author's have addressed comments from my prior review. I have no further comments

Validity of the findings

The author's have addressed comments from my prior review. I have no further comments

Reviewer 3 ·

Basic reporting

The paper is well written and pleasant to read. The state of the art is present and educational.

Experimental design

The design is well explained and simple to follow. The necessary details are present and correspond to the request of the scientific community.

Validity of the findings

Results and conclusion are strong and accurate

Additional comments

When reading the answers to all the reviewers, we can see that the authors have done a lot of work.